# Phase II randomised control feasibility trial of a nutrition and physical activity intervention after radical prostatectomy for prostate cancer

Lucy E Hackshaw-McGeagh ![ORCID],[1] Chris Penfold ![ORCID],[2] Ellie Shingler,[1] Luke A Robles,[3,4] Claire M Perks,[5] Jeff M P Holly,[5] Edward Rowe,[6] Anthony Koupparis,[6] Amit Bahl,[7] Raj Persad,[6] Constance Shiridzinomwa,[8] Lyndsey Johnson,[8] Kalina M Biernacka,[5] Aleksandra Frankow,[5] Jayne V Woodside,[9] Sarah Gilchrist,[9] Jon Oxley,[10] Paul Abrams,[6] J Athene Lane,[3] Richard M Martin[3]

For numbered affiliations see end of article.

**Correspondence to**
Dr J Athene Lane;
athene.lane@bristol.ac.uk

## ABSTRACT

**Objective** Dietary factors and physical activity may alter prostate cancer progression. We explored the feasibility of lifestyle interventions following radical prostatectomy for localised prostate cancer.

**Design** Patients were recruited into a presurgical observational cohort; following radical prostatectomy, they were offered randomisation into a 2×3 factorial randomised controlled trial (RCT).

**Setting** A single National Health Service trust in the South West of England, UK.

**Participants** Those with localised prostate cancer and listed for radical prostatectomy were invited to participate.

**Randomisation** Random allocation was performed by the Bristol Randomised Trial Collaboration via an online system.

**Interventions** Men were randomised into both a modified nutrition group (either increased vegetable and fruit, and reduced dairy milk; or lycopene supplementation; or control) and a physical activity group (brisk walking or control) for 6 months.

**Blinding** Only the trial statistician was blind to allocations.

**Primary outcome measures** Primary outcomes were measures of feasibility: randomisation rates and intervention adherence at 6 months. Collected at trial baseline, three and six months, with daily adherence reported throughout. Our intended adherence rate was 75% or above, the threshold for acceptable adherence was 90%.

**Results** 108 men entered the presurgical cohort, and 81 were randomised into the postsurgical RCT (randomisation rate: 93.1%) and 75 completed the trial. Of 25 men in the nutrition intervention, 10 (40.0%; 95% CI 23.4% to 59.3%) adhered to the fruit and vegetable recommendations and 18 (72.0%; 95% CI 52.4% to 85.7%) to reduced dairy intake. Adherence to lycopene (n=28), was 78.6% (95% CI 60.5% to 89.8%), while 21/39 adhered to the walking intervention (53.8%; 95% CI 38.6% to 68.4%). Most men were followed up at 6 months (75/81; 92.6%). Three 'possibly related' adverse events were indigestion, abdominal bloating and knee pain.

---

### Strengths and limitations of this study

► Robust, gold standard, randomised control trial 2×3 factorial design, with 6 month follow-up.
► Excellent retention rates, with only six participants withdrawing or being lost to follow-up.
► Outcomes included objective biochemical measures that are not open to responder bias.
► Except for the statistician, all researchers and participants were unblinded; blinding was not feasible given the nature of the intervention.

---

**Conclusions** Interventions were deemed feasible, with high randomisation rates and generally good adherence. A definitive RCT is proposed.

**Trial registration number** ISRCTN 99048944.

## INTRODUCTION

Prostate cancer is the most common male cancer in the UK. There were 47 740 new cases diagnosed in 2016,[1] accounting for 26% of UK male cancers. Furthermore, in 2016, there were 11 600 prostate cancer deaths, making the disease the second most common cause of UK cancer-related death.[1] Following improvements in clinical diagnosis and treatment leading to rising survival rates, more men are living longer with the disease, and there is growing interest in developing lifestyle modifications for tertiary prevention of prostate cancer morbidity and mortality.[2]

Evidence summarised by the World Cancer Research Fund[3] suggests that vegetables and fruit, and associated micronutrients (eg, carotenoids, some vitamins and flavonoids), may reduce prostate cancer risk, while calcium and dairy intake may increase

BMJ

risk. Our recent review of 44 randomised controlled trials (RCTs) of dietary, nutrition and physical activity interventions suggest various dietary interventions (eg, low fat diet, soya and soya foods, selenium, lycopene or green tea) may reduce prostate cancer progression and mortality.[2] However, many studies were assessed as being of low quality (with high or unclear risk of bias), underpowered, inadequately reported, of short duration or measuring surrogate outcomes, such as prostate-specific antigen doubling time, which has an uncertain relationship with prostate cancer progression. We therefore wanted to conduct a high-quality, low risk of bias, adequately powered, well reported, 6-month trial to explore the feasibility of a simple, easy to implement 'real life', dietary and physical activity intervention.

Lycopene, an antioxidant phytonutrient, has been observed to reduce prostate cancer progression,[4 5] while high-fat milk intake has been positively associated with prostate cancer progression.[6] However, robust RCT evidence for the effects of lycopene or milk intake in relation to prostate cancer progression is lacking.[3]

Moderate to vigorous physical activity is another potentially promising intervention[7]; research suggests that it is safe and can benefit quality of life throughout the cancer journey,[8] but the evidence is also largely based on observational data: further RCTs are needed for definitive evidence. Moderate to vigorous intensity activity is defined as activities that cause sweating and increased heart and respiratory rate, of which brisk walking is an example.[9] Brisk walking has been associated with anti-cancer cellular behaviour[10] and does not require specialised equipment or training.

Some men spontaneously and positively alter lifestyle behaviours on receiving a diagnosis of prostate cancer. However, for many men, this is not the case,[11–13] and additional interventions are necessary to promote and maintain behavioural change.[14–16] Radical prostatectomy (surgical removal of the prostate) is a common treatment for localised disease.[1] We established a 2×3 factorial RCT—the Prostate Cancer: Evidence of Exercise and Nutrition Trial (PrEvENT)—to explore the feasibility of introducing modified nutrition (increased vegetables and fruit combined with reduced dairy milk) or lycopene supplements and physical activity (brisk walking) interventions in men treated with radical prostatectomy for localised prostate cancer. Here we report the primary trial findings.

## MATERIALS AND METHODS
### Trial design and participants
The trial protocol has been published elsewhere.[17] In brief, men from a single National Health Service trust in the South West of England, UK, who were diagnosed with localised prostate cancer and listed for radical prostatectomy, were invited to participate in PrEvENT between August 2014 and May 2016. There were two phases: (1) recruitment of men scheduled for radical prostatectomy into a baseline cohort for epidemiological characterisation (ie, presurgical questionnaire data and blood sampling) of the target population; and (2) recruitment of the men once they had their radical prostatectomy into a 2×3 factorial RCT.

Cohort and RCT inclusion criteria were: localised prostate cancer; undergoing radical prostatectomy; capacity to provide informed consent; aged 18 years or over; and sufficient understanding of the English language. Exclusion criteria were: inability to give informed consent; unavailability for follow-up; identified as unsuitable to participate by the treating clinician; comorbidities (this could include uncontrolled congestive heart failure or angina, recent myocardial infarction or breathing difficulties requiring oxygen use or hospitalisation), allergies or religious beliefs that could prevent participation in the intervention (RCT only); and regularly taking lycopene supplements or routinely exercising vigorously (RCT only).

Men were approached to participate in the cohort by their clinical care team or a research nurse. Nurse-led consent at cohort recruitment included optional approval for prostate tissue extracted during surgery to be used for research purposes and willingness to be contacted about future involvement in further research. Approximately 6 weeks after surgery, the men who were participating in the cohort were invited to participate in the RCT. Six weeks postsurgery was identified as the ideal time to approach men to participate in the trial, as this is the time point where men are generally informed by their clinical team that they should be able to begin exercising again after radical prostatectomy.

### Randomisation
Random allocation was performed by the Bristol Randomised Trial Collaboration via an online system to ensure that the recruiting nurses could not uncover allocation of trial group in advance (concealment of allocation). The research nurse, research team and men were not blind to intervention allocation, because this would have been unfeasible given the nature of the interventions, while the trial statistician was blind to allocations. Men were randomised at the trial baseline appointment, where they were provided with all information related to their intervention group in person by the research nurse.

### Interventions
Men were randomised to both a physical activity intervention (two levels; brisk walking vs control) and a nutrition intervention (three levels; modified nutrition group vs lycopene vs control). Men in the physical activity intervention group were instructed to walk at a brisk pace for 30 min on at least 5 days a week in addition to their usual physical activity, while control group men were asked to continue with their usual physical activity. Men in the plant-based diet group (modified nutrition) were instructed to eat as many portions of fruit and vegetables as possible a day, aiming for at least five daily portions

(printed instructions explained how to measure a portion). In addition, men in the plant-based diet group were asked to reduce their dairy milk intake as much as possible and to use a non-dairy alternative, for example, soy, almond or rice milk as often as possible. Men in the lycopene group were asked to consume one 10 mg lycopene capsule daily (Holland and Barrett, a UK supplier of nutritional supplements, supplied to the men and paid for by the trial). Men in the nutrition control group were asked to continue with their usual diet. Control men who asked for physical activity or nutrition advice were provided with publicly available information. Instructions on how to perform the interventions were administered by the research nurse. All men were asked to follow their allocated interventions for 6 months and were contacted at 1, 2, 5, 8, 13, 15 and 18 weeks postrandomisation. The method of contact was chosen by the man: text message, emails, phone call or post. The content of the communication was structured and consistent, regardless of the method of delivery, to ensure similar contact and information provision across participants. These communications contained recipe ideas (plant-based diet group) and theory developed motivational messages to encourage continuation of the trial group intervention.[17]

### Data collection

Men attended the hospital for a research nurse appointment at trial baseline (randomisation), 3 and 6 months and completed patient-reported outcome measures (PROMs) (paper questionnaires). If the man was unable to attend the hospital, the questionnaire was sent in the post. Questionnaires analysed here to illustrate the impact of adherence were the Recent Physical Activity Questionnaire (RPAQ)[18] and Food Frequency Questionnaire (FFQ),[19] while the other PROMs are detailed elsewhere.[17] At baseline, men reported sociodemographic information (level of education, marital status, occupation and ethnicity) and family medical history of cancer. At baseline, 3 and 6 months, smoking and alcohol use, and nurse-measured anthropometry, were recorded. Non-fasting blood samples were collected at baseline and 6 months. All men completed a paper daily monitoring form for the 6-month duration logging daily step count; additionally, the physical activity intervention group recorded the duration of brisk walking in minutes; the plant-based diet nutrition group recorded the daily number of portions of fruit or vegetables consumed and the amount of dairy milk replaced by non-dairy alternatives (all/some/none), while the lycopene supplementation group recorded whether they had taken the lycopene supplement (yes/no).

### Public and patient involvement (PPI)

A prostate cancer PPI group was involved from the early protocol development stages. The PPI group contributed to discussions regarding intervention content and length, reviewed trial documentations, such as consent forms and participant information sheets, revised intervention

instructions to improve readability and completed questionnaires to assess participant burden.

### Sample size calculation

The sample size calculation, described previously,[17] identified a required minimum sample size of 75 men to estimate an adherence rate (the primary outcome, see Primary Outcomes) of 75% with a CI of ±5% (90% power, p=0.05) in each group. We aimed to recruit at least 100 men to the cohort study, anticipating an 80% randomisation rate.

### Statistical analysis

Analyses included all participants with measurements for the primary outcomes and trial baseline values at randomisation in an intention to treat analysis. All statistical analyses used Stata V.14.[20]

#### Primary outcomes

The primary outcomes were: (1) the randomisation rate (the number of men agreeing to be randomised into the RCT divided by the total number of men in the cohort) our intended rate was 80% and (2) adherence to the man's allocated intervention at 6 months following randomisation. Our intended adherence rate was 75% or above. Adherence was based on the 4 weeks prior to the participants' latest entry in their daily monitoring form. Specifically, for each intervention, to assess adherence to the plant-based diet, we analysed the proportion of days that the men reported they consumed five or more portions of fruit or vegetables and the proportion of days they replaced all or some of their dairy milk with non-dairy alternative. Adherence to the lycopene intervention was based on the proportion of days that the men reported they took the lycopene supplement. The threshold for acceptable adherence was 90%, which equates to adhering to the fruit and vegetable or lycopene interventions on 26 out of a possible 28 days. Adherence to the physical activity intervention was based on number of days of doing 30 or more minutes of self-reported brisk walking. Across the 28 days over which adherence was monitored, men were considered to have adhered 100% to the physical activity intervention if they did 30+ min of walking on at least 20 days, while 90% adherence was defined as doing 30+ min of walking on 18 of the 28 days.

#### Secondary outcomes

PrEvENT secondary outcomes were trial retention, blood biomarkers of fruit and vegetable consumption, self-reported food consumption (measured using the FFQ), self-reported physical activity (measured using the RPAQ) and trial tolerability (measured by qualitative interviews and adverse events); these outcomes are reported here. Other secondary outcomes were change in prostate specific antigen and insulin-like growth factor, change in urinary symptoms, psychological factors, health beliefs, quality of life, fatigue, general health, accelerometer and pedometer data and lifestyle behaviours (smoking

and drinking). These latter secondary outcomes are not reported here.

As secondary measures of adherence in the dietary interventions, serum concentrations of lutein, zeaxanthin, β-cryptoxanthin, α-carotene, β-carotene and lycopene were measured by reverse-phase high-performance liquid chromatography.[21] Assays were standardised against appropriate National Institute of Standards and Technology reference materials.[22] These assays are also externally quality assured by participation in the French Society for Vitamins and Biofactors quality assurance scheme.[23]

Mean portions of fruit and vegetables consumed per day were calculated from the FFQ. Physical activity energy expenditure was derived from the RPAQ. The effects of the nutrition and physical activity interventions on fruit and vegetable consumption and energy expenditure, respectively, were assessed using multivariable linear regression. As prespecified, we assumed no interaction between the interventions and tested this through repeating the multivariable linear regression analyses with an interaction term between the interventions and application of the Wald test with the null hypothesis that the interaction terms were zero. We also compared the consumption of non-dairy milk recorded in the FFQ (categorised into 'Dairy': full cream, semi skimmed, skimmed or dried milk; and 'Non-dairy': soya and other) and blood biomarkers of fruit and vegetable consumption (lutein, zeaxanthin, beta cryptoxanthin, alpha carotene, beta carotene and lycopene) between trial groups using $\chi^2$ tests for categorical outcomes and Kruskal-Wallis tests for continuous outcomes, respectively. A Bonferroni correction was applied to p values (multiplied by two) for post hoc pairwise comparisons between the nutrition groups (no comparison was made between the lycopene and plant-based interventions) calculated using pairwise $\chi^2$ tests for categorical outcomes and Wilcoxon rank-sum tests for continuous outcomes. Intervention tolerability was assessed via adverse events and post-trial qualitative interviews (reported elsewhere).

### Exploratory outcomes
To describe change in adherence from the beginning to the end of the trial period, adherence was also calculated for the first 4 weeks of daily data monitoring forms after the participants earliest entry. We also calculated weekly adherence to the interventions from 1 to 26 weeks and overlaid the postrandomisation contact reminders.

## RESULTS
### Study recruitment, participant characteristics, randomisation and follow-up
Between August 2014 and May 2016, 292 eligible men scheduled for radical prostatectomy were invited to participate and of these, 108 (37.0 %) entered the cohort study (figure 1). The most common reasons for not enrolling were: distance to travel (n=75), not being contactable (n=58) or uninterested in taking part (n=28). Of the

108 men, 96 completed cohort follow-up, a further nine were no longer eligible for the RCT, thus 87 were invited into the RCT and 81 (93.1%, 95% CI 85.8% to 96.8%) consented to be randomised, a trial primary endpoint.

Men were randomised on average around 8 weeks after surgery (range: 4 weeks (31 days) to 15 weeks (108 days)). The men were on average 64.0 years of age (SD=6.3 years), with 88% (n=68) self-reporting as white British and with a median preoperative prostate specific antigen of 8.7 ng/mL (IQR: 5.9–12.3). Almost all men were former or never smokers (97.4%, n=75), and 20% (n=16) self-reported as consuming more than 14 units of alcohol per week. In general, baseline characteristics were similarly distributed between trial groups (table 1). Seventy-five of the 81 men completed the trial (92.6 %); three were lost to follow-up and three withdrew. Those who withdrew were for reasons unrelated to the interventions. We present the results of the 75 men who completed the trial.

### Primary outcomes

#### Adherence to interventions by self-reported daily records
Adherence for at least 90% of the time was 40.0% (95% CI 23.4% to 59.3%, n=10 of 25) in the plant-based diet group for the fruit and vegetable component and 72.0% (95% CI 52.4% to 85.7%, n=18 of 25) for the dairy milk component. In the lycopene group, 78.6% (95% CI 60.5% to 89.8%, n=22 of 28) of men adhered at least 90% of the time. Adherence to the physical activity intervention was 53.8% (95% CI 38.6% to 68.4%, n=21 of 39) at least 90% of the time (table 2). Weekly adherence to the interventions appeared stable for most of the intervention period, with a potential drop-off after week 22 in the lycopene and plant based diet arms of the nutrition interventions (figure 2). Postrandomisation reminders (weeks 1, 2, 5, 8, 13 and 18) did not appear to lead to changes in adherence to the interventions.

### Secondary outcomes

#### Effects of the interventions on self-reported diet and physical activity PROMs
Those men in the plant-based diet intervention group reported consuming a mean of 9.9 portions (SD=4.7 portions) of fruit and vegetables a day based on the FFQ PROMs. This was almost four portions (adjusted difference=3.7, 95% CI 1.1 to 6.4 portions/day) more than those in the control group (mean=6.2, SD=3.8 portions) (p=0.007). Those in the lycopene intervention groups reported consuming a mean of 8.2 portions per day (SD=4.8 portions), which was 2.1 portions (95% CI –0.5 to 4.7) more per day than the control group (p=0.12) (table 3). Consumption of non-dairy milk was highest in the plant-based diet intervention group (78.3%, 95% CI 58.1% to 90.3%, n=18 of 23) and was very low otherwise (7.7%, 95% CI 2.1% to 24.1%, 2 of 26; and 0%, 95% CI 0.0% to 16.1%, 0 of 20 for the lycopene intervention and control groups, respectively, p<0.001) (table 3).

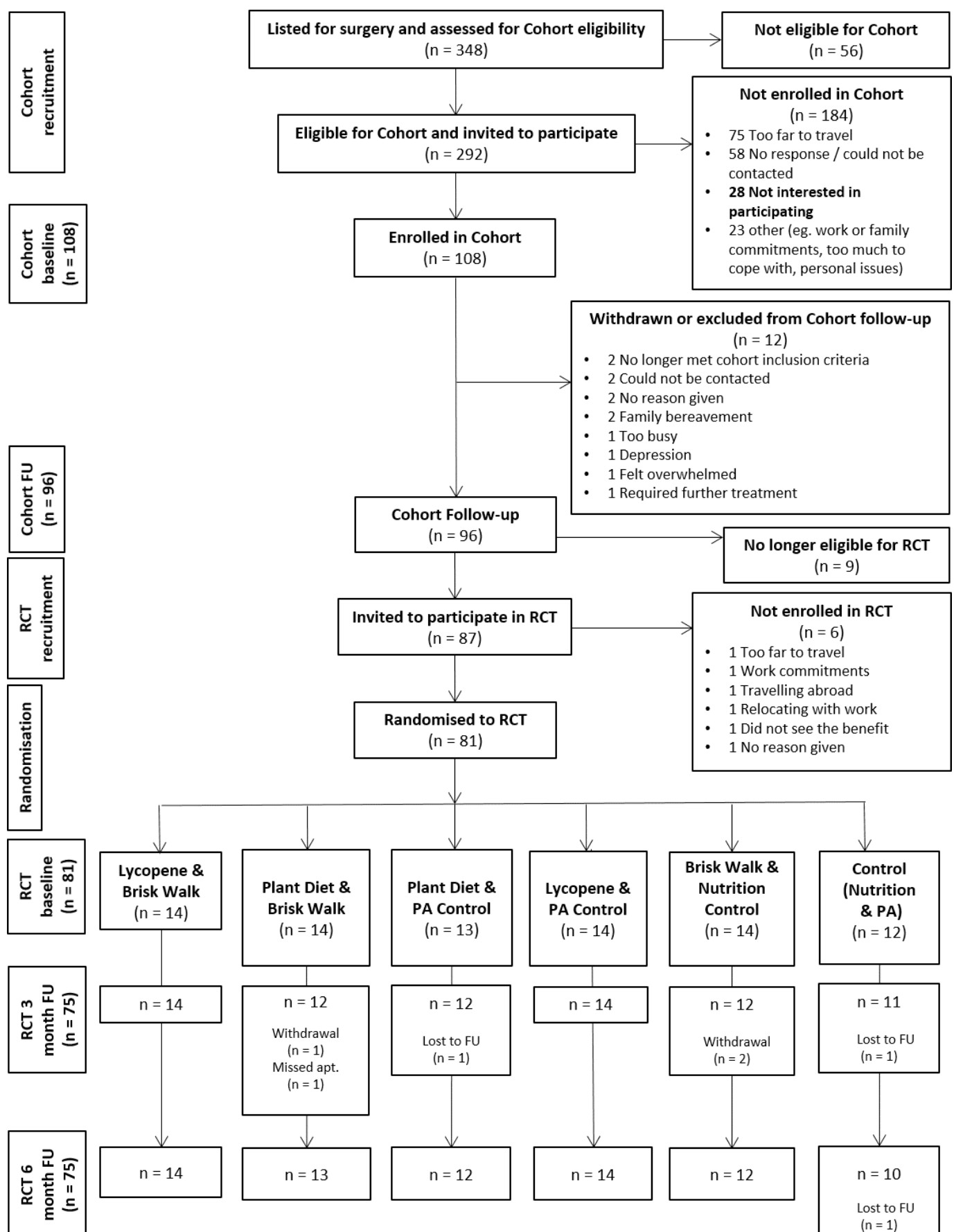

**Figure 1**  CONSORT diagram. CONSORT, Consolidated Standards of Reporting Trials; FU, follow-up; PA, physical activity; RCT, randomised controlled trial.

**Table 1** Participant characteristics, baseline measures and clinical data by intervention groups

| | n (% of participants in group) | | | | |
| --- | --- | --- | --- | --- | --- |
| | **Nutrition (n=81)** | | | **Physical activity (PA) (n=81)** | |
| | **Lycopene intervention (n=28, 34.6%)** | **Plant-based diet intervention (n=27, 33.3%)** | **Nutrition control (n=26, 32.1%)** | **PA control (n=39, 48.1%)** | **PA intervention (n=42, 51.9%)** |
| **Participant characteristics** | | | | | |
| Age (years)* | 62.9 (7.7) | 64.6 (5.7) | 64.7 (5.5) | 62.5 (6.9) | 65.5 (5.5) |
| **Ethnicity** | | | | | |
| White British | 23 (88.5) | 23 (85.2) | 22 (91.7) | 33 (86.8) | 35 (89.7) |
| **Highest education level†** | | | | | |
| Standard or less (eg, o-levels and GCSE) | 7 (26.9) | 12 (44.4) | 14 (60.9) | 20 (52.6) | 13 (34.2) |
| Further education (eg, A-levels, HND and university degree) | 19 (73.1) | 15 (55.6) | 9 (39.1) | 18 (47.4) | 25 (65.8) |
| **Marital status** | | | | | |
| Married/in a relationship | 23 (88.5) | 22 (81.5) | 22 (95.7) | 33 (86.8) | 34 (89.5) |
| **Father had prostate cancer** | | | | | |
| Yes | 11 (44.0) | 2 (7.4) | 1 (4.3) | 8 (21.6) | 6 (15.8) |
| **Smoking status** | | | | | |
| Current | 0 (0.0) | 0 (0.0) | 2 (7.7) | 1 (2.7) | 1 (2.5) |
| Former | 9 (37.5) | 13 (48.1) | 16 (61.5) | 21 (56.8) | 17 (42.5) |
| Never | 15 (62.5) | 14 (51.9) | 8 (30.8) | 15 (40.5) | 22 (55.0) |
| Alcohol consumption (units/week)‡ | 1.0 (0.0; 7.0) | 4.0 (0.0; 12.0) | 4.5 (0.0; 22.0) | 2.0 (0.0; 7.0) | 4.0 (0.0; 13.0) |
| **Alcohol consumption (categorised)§** | | | | | |
| Non-drinker (0 units/week) | 14 (50.0) | 9 (33.3) | 8 (30.8) | 17 (43.6) | 14 (33.3) |
| ≤14 units/week | 11 (39.3) | 14 (51.9) | 9 (34.6) | 15 (38.5) | 19 (45.2) |
| >14 units/week | 3 (10.7) | 4 (14.8) | 9 (34.6) | 7 (17.9) | 9 (21.4) |
| Body mass index (kg/m$^2$)* | 26.2 (3.6) | 26.7 (3.6) | 26.8 (3.1) | 26.6 (3.5) | 26.6 (3.3) |
| **Baseline measures and clinical data** | | | | | |
| Presurgical prostate specific antigen (ng/mL)‡ | 7.8 (5.2; 11.0) | 8.4 (5.7; 9.6) | 10.6 (8.4; 15.0) | 8.4 (5.7; 11.0) | 8.9 (6.4; 12.7) |
| Baseline fruit and vegetable consumption (portions/day)* | 7.5 (4.7) | 8.5 (5.7) | 7.3 (2.4) | 8.2 (5.2) | 7.4 (3.7) |
| Baseline energy expenditure (kJ/kg/day)* | 27.2 (32.9) | 22.8 (13.9) | 27.4 (21.6) | 23.1 (23.0) | 28.3 (25.0) |
| **Type of milk consumed most often** | | | | | |
| Dairy | 22 (88.0) | 22 (84.6) | 24 (96.0) | 30 (83.3) | 38 (95.0) |
| Non-dairy | 3 (12.0) | 4 (15.4) | 1 (4.0) | 6 (16.7) | 2 (5.0) |
| Lutein – µmol/L‡ | 0.36 (0.28; 0.55) | 0.40 (0.32; 0.52) | 0.35 (0.20; 0.46) | 0.35 (0.28; 0.52) | 0.38 (0.28; 0.51) |
| Zeaxanthin – µmol/L‡ | 0.11 (0.09; 0.16) | 0.10 (0.09; 0.12) | 0.10 (0.06; 0.13) | 0.11 (0.08; 0.12) | 0.10 (0.08; 0.14) |
| Beta cryptoxanthin – µmol/L‡ | 0.22 (0.16; 0.36) | 0.25 (0.15; 0.36) | 0.19 (0.11; 0.39) | 0.22 (0.14; 0.32) | 0.24 (0.14; 0.40) |
| Alpha carotene – µmol/L‡ | 0.07 (0.05; 0.09) | 0.08 (0.06; 0.10) | 0.07 (0.05; 0.08) | 0.07 (0.05; 0.08) | 0.07 (0.06; 0.09) |
| Beta carotene – µmol/L‡ | 0.64 (0.46; 1.06) | 0.72 (0.50; 1.16) | 0.56 (0.47; 0.84) | 0.66 (0.47; 0.89) | 0.64 (0.48; 1.07) |
| Lycopene – µmol/L‡ | 1.35 (1.03; 1.69) | 1.24 (0.92; 1.44) | 1.20 (0.90; 1.59) | 1.23 (0.92; 1.47) | 1.29 (1.04; 1.62) |

*Values are mean (SD).
†O-level, GCSE: national school examinations at age 16 years. A-level: national school examinations at age 18 years. Higher national diploma (HND) is a work-related course provided by higher and further education colleges in the UK. A full-time HND takes 2 years to complete and generally is the equivalent to 2 years at university.
‡Values are median (25%; 75%).
§Non-drinker: 0 units/week; moderate: >0 and ≤14 units/week; hazardous: >14 and ≤50 units/week; harmful: >50 units/week.

Those men in the physical activity intervention and control groups reported very similar mean energy expenditure based on the PROMs data (mean=39.7 kJ/kg/day, SD=26.3 kJ/kg/day, and mean=40.8 kJ/kg/day, SD=28.6 kJ/kg/day, respectively) (adjusted difference=−0.8 kJ/kg/day, 95% CI −13.6 to 11.9 kJ/kg/day, p=0.90) (table 3).

**Table 2** Adherence to the interventions by self-reported daily monitoring*

| | Adherence to lycopene intervention* | Adherence to plant-based diet intervention* | | Adherence to physical activity intervention* |
| --- | --- | --- | --- | --- |
| | | Fruit and vegetables | Dairy replacement | |
| **Physical activity (PA)** | | | | |
| PA control (n=36) | 30.6% (n=11) | 13.9% (n=5) | 30.6% (n=11) | – |
| PA intervention (n=39) | 28.2% (n=11) | 12.8% (n=5) | 18.0% (n=7) | 53.8% (n=21) |
| **Nutrition** | | | | |
| Lycopene intervention (n=28) | 78.6% (n=22) | – | – | 28.6% (n=8) |
| Plant-based diet intervention (n=25) | – | 40.0% (n=10) | 72.0% (n=18) | 28.0% (n=7) |
| Nutrition control (n=22) | – | – | – | 27.3% (n=6) |

*Adherence was based on the 4 weeks prior to the participants' latest entry in their daily monitoring form. The threshold for acceptable adherence was 90%, which equates to adhering to the fruit and vegetable or lycopene interventions on 26 out of a possible 28 days. Adherence to the PA intervention equates to doing 30+ min of walking on 18 of the 28 days.

There was no evidence of an interaction between the nutrition and physical activity interventions (F=0.65, p=0.63) so the data are not shown here.

### Effects of the nutrition intervention on blood biomarkers of fruit and vegetable consumption

Lutein (p=0.07), beta cryptoxanthin (p=0.07), beta carotene (p=0.09) and lycopene (p=0.07) were higher in the plant-based intervention group than the control group. There was no difference in lycopene between the lycopene intervention group compared with the control group (p=0.64). No other differences were found between the blood biomarkers (table 4).

### Intervention tolerability

Nine adverse events, self-reported by participants, were classed as unrelated to the interventions and included fever, alcohol withdrawal symptoms, urinary symptoms, diverticulitis, pneumonia, *Campylobacter* infection and hernia repair procedure. Three adverse events were considered as 'possibly related to the interventions'; these were indigestion, abdominal bloating and knee pain. Data from qualitative interviews reported elsewhere also suggested that men found all the interventions to be acceptable, easy to accommodate and well tolerated.[24]

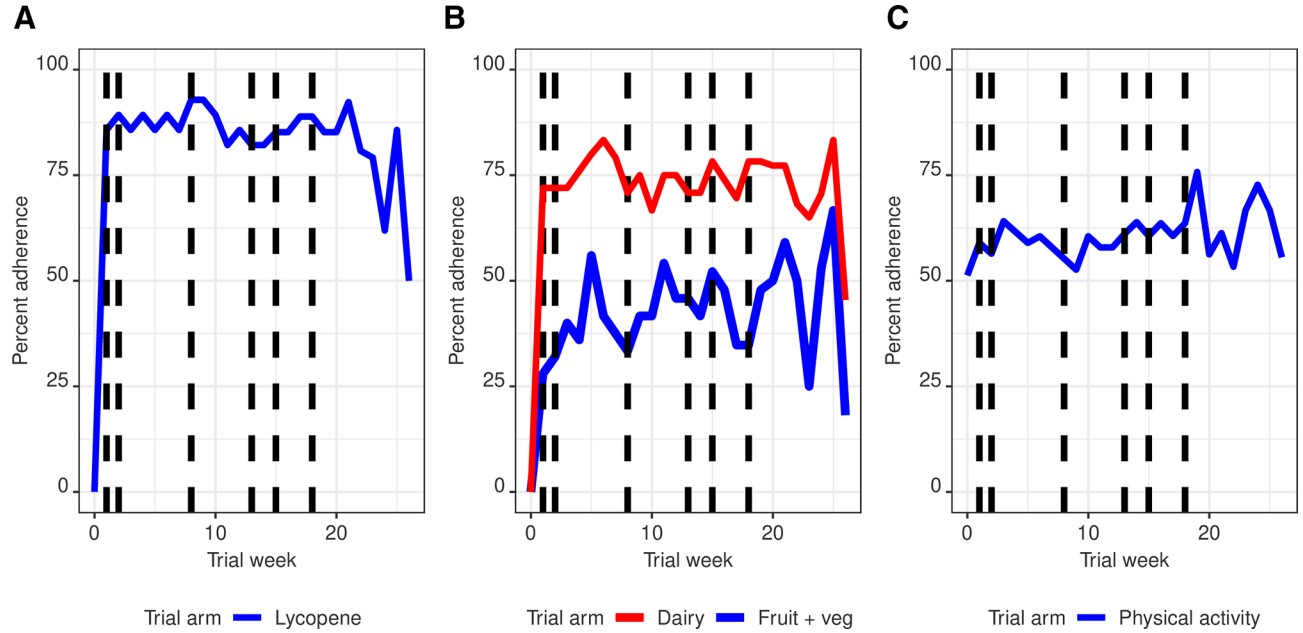

**Figure 2** Weekly adherence to the intervention from 0 to 26 weeks for participants randomised to the lycopene (A) or plant-based diet (B) arms of the nutrition intervention and physical activity intervention arm (C). Dashed vertical lines represent timepoints (weeks 1, 2, 8, 13, 15 and 18) at which reminders were sent to participants.

**Table 3** Effects of interventions on fruit and vegetable consumption, non-dairy milk consumption and energy expenditure at 6 months

| | Nutrition (n=75) | | | Physical activity (n=75) | |
|---|---|---|---|---|---|
| | Lycopene intervention (n=28) | Plant-based diet intervention (n=25) | Nutrition control (n=22) | Physical activity control (n=36) | Physical activity intervention (n=39) |
| Fruit and vegetable (portions/day), mean (SD) | 8.2 (4.8) | 9.9 (4.7) | 6.2 (3.8) | 8.3 (4.9) | 8.1 (4.5) |
| Adjusted difference in fruit and vegetable consumption (portions/day) compared with the control group* (95% CI) | 2.07 (-0.51 to 4.65) | 3.73 (1.07 to 6.38) | – | – | −0.20 (−2.37 to 1.98) |
| P value | 0.12† | 0.007† | | | 0.86† |
| Energy expenditure (kJ/kg/day), mean (SD) | 41.8 (23.7) | 43.1 (32.0) | 34.9 (26.1) | 40.8 (28.6) | 39.7 (26.3) |
| Adjusted difference in energy expenditure (kJ/kg/day) compared with the control group* (95% CI) | 6.91 (−8.77 to 22.6) | 8.19 (−7.89 to 24.3) | – | – | −0.84 (−13.6 to 11.9) |
| P value | 0.38† | 0.31† | | | 0.90† |
| Type of milk consumed most often | | | | | |
| Dairy, n (%) | 24 (92.3) | 5 (21.7) | 20 (100.0) | 22 (66.7) | 27 (75.0) |
| Non-dairy, n (%) | 2 (7.7) | 18 (78.3) | 0 (0.0) | 11 (33.3) | 9 (25.0) |
| P values | 0.41‡ | <0.001‡ | <0.001§ | | 0.08§ |

*Fruit and vegetable consumption adjusted for physical activity intervention group and energy expenditure adjusted for nutrition intervention group.
†P values from multivariate linear regression.
‡P values from Wilcoxon sign-rank test with Bonferroni correction.
§P values from Wilcoxon sign-rank tests.

### Exploratory analysis

Change in adherence from the beginning to the end of the trial period was examined. In the plant-based group, there was an increase in adherence at 6 months compared with the start of the intervention, both for the fruit and vegetable consumption, which increased from 20.0% (95% CI 8.9% to 39.1%, n=5 of 25), and the reduced dairy consumption, which increased from 60.0% (95% CI 40.7% to 76.6%, n=15). Adherence in the lycopene group fell slightly from 89.3% (95% CI 72.8% to 96.3%, n=25 of 28) within the first 4 weeks and in the physical activity group, which fell slightly from 59.0% (95% CI 43.4% to 72.9%, n=23 of 39) during the first 4 weeks to reported adherence at 6-month follow-up (table 5).

### DISCUSSION

PrEvENT recruited 37.0% of eligible men into the cohort and randomised 93.1% of men from the cohort into the RCT. This is higher than our anticipated randomisation rate of 80%. Forty per cent of men in the plant-based diet group adhered to the fruit and vegetable component of the intervention at 6 months, with more than 70% adhering to the dairy replacement component. Almost 80% of men in the lycopene group adhered to the intervention at 6 months and just under 54% of men adhered to the physical activity intervention at 6 months. Those in the plant-based diet consumed 9.9 portions of fruit and vegetables per day, which was more than the nutrition control (6.4 portions). The lycopene group consumed 8.2 portions per day. Over 90% completed 6-month follow-up.

**Table 4** Effects of interventions on blood biomarkers at 6 months

| | Nutrition (n=75) | | | | P (pairwise comparisons)† | | Physical activity (n=75) | | |
| | Lycopene intervention (n=28) Median (IQR) | Plant-based diet intervention (n=25) Median (IQR) | Nutrition control (n=22) Median (IQR) | P value* | Lycopene versus control | Plant based versus control | Physical activity intervention (n=39) Median (IQR) | Physical activity control (n=36) Median (IQR) | P value* |
|---|---|---|---|---|---|---|---|---|---|
| Lycopene (µmol/L) | 1.30 (0.96; 1.60) | 1.41 (1.12; 1.79) | 1.16 (0.83; 1.39) | 0.11 | 0.64 | 0.07 | 1.27 (0.95; 1.71) | 1.36 (0.97; 1.58) | 0.90 |
| Lutein (µmol/L) | 0.37 (0.28; 0.46) | 0.49 (0.36; 0.57) | 0.36 (0.27; 0.48) | 0.05 | 1.00 | 0.07 | 0.40 (0.29; 0.49) | 0.38 (0.26; 0.55) | 0.56 |
| Zeaxanthin (µmol/L) | 0.09 (0.07; 0.16) | 0.12 (0.10; 0.16) | 0.10 (0.08; 0.14) | 0.26 | 0.92 | 0.22 | 0.12 (0.08; 0.15) | 0.11 (0.08; 0.16) | 0.89 |
| Beta cryptoxanthin (µmol/L) | 0.22 (0.11; 0.39) | 0.30 (0.23; 0.51) | 0.23 (0.12; 0.29) | 0.06 | 1.00 | 0.07 | 0.24 (0.13; 0.50) | 0.23 (0.16; 0.35) | 0.44 |
| Alpha carotene (µmol/L) | 0.07 (0.06; 0.09) | 0.08 (0.06; 0.13) | 0.07 (0.05; 0.09) | 0.26 | 1.00 | 0.28 | 0.07 (0.05; 0.10) | 0.07 (0.06; 0.09) | 0.52 |
| Beta carotene (µmol/L) | 0.59 (0.37; 0.97) | 0.81 (0.54; 1.24) | 0.57 (0.46; 0.79) | 0.11 | 1.00 | 0.09 | 0.73 (0.45; 1.16) | 0.64 (0.43; 0.93) | 0.47 |

*P-values from Kruskal-Wallis tests unless stated otherwise.
†P values from Wilcoxon rank-sum tests, with Bonferroni correction unless stated otherwise.

Adherence for both elements of the plant-based intervention increased from the first month of the intervention to the last month. This may reflect the men's need to develop new shopping, eating and social behaviours to incorporate the increased fruit and vegetable consumption as well as the reduction in dairy milk. It should be noted that our definition of adherence was strict, expecting participants to follow the intervention more than 90% of the time. Previous diet and physical activity interventions in cancer populations have reported higher adherence rates of around 70%. However, in these instances, the interventions were either less intensive or definitions of adherence were not as strict as the current study.[25–27]

The findings from our blood biomarker analysis further contribute to the conclusion of acceptable adherence, with higher levels of lutein and beta cryptoxanthin evident in the plant-based population at 6 months, compared with the other groups. Both carotenoids are found in fruit and vegetables, for example, dark leafy greens, oranges, red peppers and carrots, thus implying higher consumption of certain fruits and vegetables in the plant-based diet group compared with the other groups. Almost 80% of the plant-based group self-reported consuming non-dairy milk products at 6 months compared with 8% of the lycopene group and none of the control group, suggestive of high compliance with that intervention.

Adherence to lycopene supplements was high at almost 80% at 6 months, a slight drop from the first 4 weeks of the trial where almost 90% adherence was reported. This may have been down to the simplicity of taking a lycopene supplement, where the majority of men were able to

**Table 5** Exploratory data: adherence to the interventions by self-reported daily monitoring (first 4 weeks)*

| | Adherence to lycopene intervention* | Adherence to plant-based diet intervention* | | Adherence to physical activity intervention* |
| | | Fruit and vegetables | Dairy replacement | |
|---|---|---|---|---|
| **Physical activity (PA)** | | | | |
| PA control (n=36) | 33.3% (n=12) | 2.8% (n=1) | 19.4% (n=7) | – |
| PA intervention (n=39) | 33.3% (n=13) | 10.3% (n=4) | 20.5% (n=8) | 59.0% (n=23) |
| **Nutrition** | | | | |
| Lycopene intervention (n=28) | 89.3% (n=25) | – | – | 39.3% (n=11) |
| Plant-based diet intervention (n=25) | – | 20.0% (n=5) | 60.0% (n=15) | 20.0% (n=5) |
| Nutrition control (n=22) | – | – | – | 31.8% (n=7) |

*Adherence was based on the first 4 weeks of daily data monitoring forms after the participants' earliest entry. The threshold for acceptable adherence was 90%, which equates to adhering to the fruit and vegetable or lycopene interventions on 26 out of a possible 28 days. Adherence to the PA intervention equates to doing 30+ min of walking on 18 of the 28 days.

easily incorporate the capsule into their daily routine.[24] It should be noted that despite excellent adherence to the lycopene supplementation, no differences in lycopene levels were reported in the blood biomarkers between men in the lycopene and other groups. Assuming self-reported lycopene supplementation consumption was correct, this finding may suggest the supplement dosage of 10 mg per day was not high enough. This should be considered when designing future lycopene interventions. Our high adherence rates, however, confirm the feasibility of this population taking a daily lycopene supplement, which was an outcome of the trial.

The plant-based adherence rates were lower than the anticipated 75%, although adherence to the reduced dairy milk component were higher, almost reaching this target; the lycopene adherence rates were higher than the proposed 75%. Almost 54% of participants adhered to the physical activity intervention at 6 months, slightly lower than the 59% adhering within the first 4 weeks of the trial.

Overall, these randomisation and adherence rates suggest that men were willing to take part and generally followed the intervention instructions well. It should be noted that as a feasibility trial, we were not concerned with the clinical relevance of these changes, but whether participants would make, and adhere to, the changes for the duration of the trial.

The finding that participants in the plant-based diet intervention group consumed four additional daily portions of fruit or vegetables than the control group is substantial from a public health perspective. The lycopene intervention group also consumed a significantly larger amount of fruit and vegetables than the control group, although less than the plant-based diet participants. This may have been for one of two reasons. First, the information sheet explained that lycopene is found in red food products such as tomatoes and grapefruit, thus the lycopene group may have increased fruit and vegetable consumption for this reason, as well as taking the supplement. Second, it may have been the case that being in an active nutrition group, regardless of the specific instructions, increased participants' awareness about their nutrition, which may have resulted in them consuming more fruit and vegetables.[12]

Overall consumption of fruit and vegetables of all participants was reported to be high. Seven or eight portions per day were reported at baseline, increasing to 8–10 portions at 6 months (nutrition control was slightly lower at six portions). This is proportionally higher than the national average. In 2013/2014, 27% of those aged 19–64 years and 35% of those aged 65 years and over met the 5-A-Day recommendation.[28 29] The average consumption for these age groups were 4 and 4.2 portions of fruit and vegetables per day, respectively.[29] There may be a number of explanations for this. Due to the information in the public domain surrounding prostate cancer, which includes general advice to eat a healthy balanced diet, many of the men may have increased their fruit and

vegetable consumption at time of diagnosis.[11 12] Alternatively, as all participants had already participated in the cohort, which asked questions about their diet, this may have made them consider their diet prior to entering the RCT and thus increased consumption.[12] Finally, there may have been some over-reporting of fruit and vegetable consumption, either due to social desirability, inconsistent recall or as a result of previously reported issues with the FFQ.[30] This may explain the disparity between the reported high consumption of fruit and vegetables but lower adherence to the plant-based intervention.

There were no differences in levels of energy expenditure reported between the physical activity intervention group and control group. It may have been the case that no differences existed or that the RPAQ was not sensitive enough to detect differences. Alternatively, it may have been that the physical activity intervention was not vigorous enough to warrant a difference in score between the physical activity intervention groups. This is something to be noted for future physical activity interventions, with the possibility of increasing the intensity of the intervention, with the potential to influence weight management.

## LIMITATIONS

There are some limitations. Except for the statistician, all researchers and participants were unblinded. Men were provided with information on the specifics of all trial groups and therefore there may have been cross over in behaviour, potentially diluting results. However, blinding was not feasible given the nature of the intervention. The population lacked heterogeneity, particularly with regards to ethnicity, possibly limiting the generalisation of results. Many of the key variables were assessed via PROMs, potentially causing responder bias. However, we also report objective biochemical measures that supported the inference of good adherence, particularly within the plant-based group. It should be noted that the two key reasons for not enrolling were having too far to travel to attend research clinics or no response to the invitation being received. If these two barriers were removed, then our data suggest that the cohort recruitment could potentially have increased to 51%.

## CONCLUSION

The PrEvENT interventions were feasible as evidenced by high randomisation rates, high retention rates and excellent tolerability. Men demonstrated relatively good adherence to all interventions, particularly the lycopene supplement, brisk walking and the reduced dairy milk element of the plant-based diet interventions. Positive effects on fruit and vegetable consumption were seen in the plant-based diet and lycopene groups. The development of a definitive RCT should consider: travel distance to research clinics; increasing the intensity of the physical activity intervention and dosage of the lycopene

supplementation; and possibly reducing the minimum criteria for adherence.

## Author affiliations
[1]National Institute for Health Research (NIHR) Biomedical Research Centre (Nutrition Theme), University of Bristol, Bristol, UK
[2]National Institute for Health Research (NIHR) Biomedical Research Centre (Surgical Innovation Theme), Musculoskeletal Research Unit, University of Bristol, Bristol, UK
[3]Bristol Medical School: Population Health Sciences, University of Bristol, Bristol, UK
[4]MRC Integrative Epidemiology Unit, University of Bristol, Bristol, UK
[5]Bristol Medical School: Translational Health Sciences, University of Bristol, Bristol, UK
[6]Bristol Urology Institute, Department of Urology, North Bristol NHS Trust, Bristol, UK
[7]Bristol Haematology and Oncology Centre, University Hospitals Bristol, Bristol, UK
[8]Clinical Research Centre, North Bristol NHS Trust, Bristol, UK
[9]Institute for Global Food Security, Queens University Belfast, Belfast, UK
[10]Department of Cellular Pathology, North Bristol NHS Trust, Bristol, UK

**Acknowledgements** The authors wish to acknowledge Mr Stu Toms for his development of the trial database and data management support throughout.

**Contributors** LEHM conceived and designed the work, acquired and interpreted data and drafted the manuscript. CP interpreted data and drafted the manuscript. ES and LAR acquired data. CMP, JMPH, ER, AK, AB and RP designed the work. CS, LJ, KMB and AF acquired data. JVW and SG interpreted data. JO and PA designed the work. JAL and RMM conceived the work, designed the work and interpreted data. All authors revised the manuscript, approved the final version of the manuscript and agree to be accountable for all aspects of the work.

**Funding** This study was supported by the NIHR Biomedical Research Centre at University Hospitals Bristol NHS Foundation Trust and the University of Bristol. The Bristol Randomised Trials Collaboration (BRTC) receives National Institute for Health Research CTU Support Funding. The BRTC is a UK Clinical Research Collaboration registered CTU and a member of the NCRI Cancer Clinical Trials Group.

**Competing interests** None declared.

**Patient consent for publication** Not required.

**Ethics approval** The trial was performed in accordance with the Declaration of Helsinki, and the ethical principles of the International Conference on Harmonisation – Good Clinical Practice. All study procedures were approved by the National Research Ethics Committee South West – Cornwall and Plymouth (REC reference number 14/SW/0056). All men provided fully informed written consent at both the cohort and RCT recruitment stages. The trial was registered with the ISRCTN registry on 17 November 2014 (ISRCTN99048944).

**Provenance and peer review** Not commissioned; externally peer reviewed.

**Data availability statement** Data are available on reasonable request.

**ORCID iDs**
Lucy E Hackshaw-McGeagh http://orcid.org/0000-0002-9226-5115
Chris Penfold http://orcid.org/0000-0001-8654-353X

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
