## [Reviewer comments · BMJ Open]

ARTICLE DETAILS

TITLE (PROVISIONAL)	PHASE II RANDOMISED CONTROL FEASIBILITY TRIAL OF A NUTRITION AND PHYSICAL ACTIVITY INTERVENTION AFTER RADICAL PROSTATECTOMY FOR PROSTATE CANCER
AUTHORS	Hackshaw-McGeagh, Lucy E.; Penfold, Chris; Shingler, Ellie; Robles, Luke; Perks, Claire; Holly, Jeff; Rowe, Edward; Koupparis, Anthony; Bahl, Amit; Persad, Raj; Shiridzinomwa, Constance; Johnson, Lyndsey; Biernacka, Kalina; Frankow, Aleksandra; Woodside, Jayne; Gilchrist, Sarah; Oxley, Jon; Abrams, Paul; Lane, Athene; Martin, Richard

VERSION 1 - REVIEW

REVIEWER	Fiona Stacey The University of Newcastle, Australia
REVIEW RETURNED	28-Feb-2019

GENERAL COMMENTS	The paper describes the results of a factorial RCT to determine the feasibility of a nutrition and physical intervention on men with localised prostate cancer who had undergone radical prostatectomy. The paper is well-written and provides a detailed report of the findings of the study. Minor comments are provided below. • In the introduction, the authors describe the lack of specific evidence for MVPA, but there is no mention of the large evidence base supporting the efficacy of physical activity for cancer survivors. Methods: • Please add details of ethics approval.• Please expand on what co-morbidities were excluded.• Who invited the cohort participants to take part in the RCT?• Did men in the lycopene and physical activity groups also receive communication during the intervention? What was the content of these communications?• The primary outcome was adherence at 6 months. This is defined as the 4 weeks prior to the most recent entry in daily monitoring form. It's not clear why this definition was chosen. Why not use an adherence rate over the whole intervention period, or track weekly adherence rates? Results • The results indicate that men in the plant-based diet intervention were consuming more fruit and vegetables than men in the control
---

	group. Are you able to determine whether this is as a result of increasing fruit or vegetables or both?  • Please report how energy expenditure was assessed? Based on the secondary outcomes measures in the method, I had expected to see self-reported physical activity behaviour reported here instead of energy expenditure. • For the exploratory analysis of change in adherence, I would like to see the results quantify the change from baseline to 6 months. • To determine the effect of the intervention communications, it would interesting to see if adherence rates were stable over the intervention period, or if adherence increased with the scheduled study communications. Additional comments:  • Reference 1 quoting prostate cancer statistics in the UK is from 2016. Is there any more recent data available? • Table 4: "Non-dairy milk consumption" is in the title but results are not reported in the table? • Is there any information on the number of patients with localised prostate cancer in the UK, and the proportion treated with prostatectomy? Is prostatectomy the recommended treatment for men diagnosed with localised prostate cancer? • What is the evidence for choosing an adherence rate of more than 90%? Has this threshold been used in other studies? • It's not clear why adherence to the fruit and vegetable intervention is low, given that consumption is relatively high. • The conclusion states that the interventions were acceptable, although this specific data is not reported in this manuscript.
--	--

REVIEWER	Ciaran Fairman Edith Cowan University, Australia
REVIEW RETURNED	30-Apr-2019

GENERAL COMMENTS	The manuscript titled "PHASE II RANDOMISED CONTROL TRIAL OF A NUTRITION AND PHYSICAL ACTIVITY INTERVENTION AFTER RADICAL PROSTATECTOMY FOR PROSTATE CANCER" investigated the adherence of men with prostate cancer to various nutrition and physical activity interventions. The authors should be commended for conducting work in an important population in men with prostate cancer surrounding surgery. Strengths of the manuscript include a 6-month follow-up, and the investigation of various combinations of physical activity and nutrition. There are some comments regarding the manuscript that if addressed, could improve enhance the interpretation of the results. Introduction Line 89 - The authors recent review identified a variety of diets that may have a favorable impact on prostate cancer progression and mortality. Perhaps somewhere here or elsewhere in the manuscript, a rationale as to why the specific plant-based intervention was chosen. For example, men with prostate cancer, particularly those who received androgen deprivation therapy, at a heightened risk of muscle loss. Consequently, an argument could be made for the investigation of protein and/or resistance training as opposed to plant-based diet and walking. Methods
---

Line 135 - It's likely that 6-weeks post-surgery was chosen to allow to for recovery from surgery prior to physical activity? With many pre-post exercise interventions in exercise oncology investigating shorter timelines of exercise in relation to surgery, or even throughout, it might be helpful to have a rationale here as to why 6-weeks post-surgery was chosen.

Lines 150-159 - was any other dietary advice offered beyond consumption of fruit and vegetables? Was calorie or protein intake discussed at all? Given the impact of PC treatment on weight and lean body mass loss, these are certainly of concern in this population. It might be worth providing more detail of the intervention here.

Further, how were men advised on the physical activity/nutrition guidelines - was this a one day event where men were given an overview. Was it group or individual discussions? At what point did the men receive this information? Any additional information here would be great for interpretation!

Lines 161-162 - It would make sense that varying methods of communication were chosen to keep in touch with participants. Was there any rationale as to which form of communication was chosen (participant preference for example), or if a combination was used with each participant. It would be reasonable to suggest that different modes of contact may have different degrees of impact vs. efficacy (i.e. 30 min phone call vs letter in the post). Some further insight into decisions made here would be helpful in the interpretation of this intervention.

Participant Characteristics

- Do the authors have any further information on treatment participants received for their diagnosis? Did anyone received chemotherapy, radiation or hormonal therapy whilst participating in the intervention?

Table 1. It might be helpful to include baseline measures of nutrition and PA in the tables where effects are displayed. It is difficult to go back and forth between tables when examining baseline values vs 6-month data.

How was energy expenditure calculated? Was this extrapolated from PA questionnaires or accelerometry data? This information would be valuable to include in the methods

Could the authors estimate overall calories and macronutrient information? If a definitive RCT is being proposed from the results of this trial, this information would be critical in determining if compensatory dietary patterns were formed as a result of advice from different interventions.

Is there a reason other timepoints of data collection aren't included? This would be extremely valuable in tracking timeline of intervention effects.

Line 312 it's unclear looking at the data in the tables for increases in these micronutrients, what the clinical relevance of these small increases in intake are (e.g. 1.35 umol/l v 1.2 umol/l v 1.20 umol/l for lycopene). A discussion of this would be helpful.

	Discussion One concern regarding the interpretation of the result of this trial is regarding the effects of the plant based recommendations for increasing fruits and vegetables. All groups were consuming greater than 5 servings per day at baseline. Consequently, it's unknown if there is a benefit to increasing intake beyond this. Certainly, it might be helpful to discuss the implications of this more. Playing extreme devils advocate, one could question the value of a nutrition intervention aimed at increasing fruit and vegetable consumption if individuals were already consuming an adequate amount? The authors do a great job of offering insight as to why the PA intervention was largely ineffective at increasing energy expenditure. What might be helpful to discuss would be 1) why EE is chosen as the outcome for the physical activity intervention - Is this following the line that increases in EE could be helpful in weight management? If so, in addition to discussing the potential need for more intense PA interventions, could be the discussion of interventions that would be targeting to clinically relevant outcomes in this patient population (loss of lean body mass, loss of bone mineral density and accumulation of fat mass from various treatments). Certainly, it appears as though a lifestyle intervention aimed at weight management rather than PA or EE alone would be valuable here. A primary concern reading this manuscript is the adherence to various physical activity and nutrition interventions as an outcome (as they relate to the specific interventions selected). For example, if a definitive RCT is proposed from the results of this trial - is an intervention aimed at increasing brisk walking and fruit and vegetable consumption sufficient to alter clinically relevant outcomes? Certainly, it appears as though both the PA and nutrition interventions, whilst adherence was high, didn't have a meaningful impact PA levels, EE, or nutritional intake (beyond a slight increase in fruit and vegetable intake) It might be helpful to have a discussion surrounding how future research might build on this trial. For example, the physiological and psychosocial burden of prostate cancer treatments are well documented in the literature. Further, there is a wide variety of exercise and nutrition interventions aimed at either weight management or lean body mass, bone health or other outcomes impacted by cancer treatment. It is unclear how expanding this intervention or moving forward with a definitive RCT would lead to improvements in health-related outcomes in this population.
--	---

VERSION 1 – AUTHOR RESPONSE

Reviewer: 1

Introduction:

- In the introduction, the authors describe the lack of specific evidence for MVPA, but there is no mention of the large evidence base supporting the efficacy of physical activity for cancer survivors.

We acknowledge this omission and have added the following text and additional reference.

“Moderate-vigorous physical activity is another potentially promising intervention (7); evidence suggests that it is safe and can benefit quality of life throughout the cancer journey (8) but the evidence is also largely based on observational data: further RCTs are needed for definitive evidence” (line 104)

Methods:

- Please add details of ethics approval.

Information relating to ethical approval is located within the ‘Additional Information’ section, under the sub-heading ‘Ethics approval and consent to participate’, (line 472), as requested by the journal guidelines

- Please expand on what co-morbidities were excluded.

The trial protocol does not define specific co-morbidities which would automatically result in exclusion from the trial. The definition is ‘Co-morbidities which could prevent participation in the intervention ie. this could include uncontrolled congestive heart failure or angina, recent myocardial infarction or breathing difficulties requiring oxygen use or hospitalisation.’ It was left to the discretion of the research nurse and participant to discuss co-morbidities. However, to provide further information for the purpose of the manuscript, the follow text has been added.

“co-morbidities (this could include uncontrolled congestive heart failure or angina, recent myocardial infarction or breathing difficulties requiring oxygen use or hospitalisation) allergies or religious beliefs which could prevent participation in the intervention (RCT only)” (line 134)

- Who invited the cohort participants to take part in the RCT?

Full details of how participants were approached and recruited can be found in the published protocol paper (referenced in the manuscript), however for clarity in the manuscript we have included the following sentence.

“Men were approached to participate in the cohort by their clinical care team, or a research nurse.” (line 140)

- Did men in the lycopene and physical activity groups also receive communication during the intervention? What was the content of these communications?

We feel that this comment has been addressed within the manuscript (line 172). An additional sentence has been added in response to a reviewers comment below. Further information is provided in the protocol paper which is referenced after the text below.

“All men were asked to follow their allocated interventions for six months and were contacted at one, two, five, eight, 13, 15 and 18 weeks post-randomisation. The method of contact was chosen by the man: text message, emails, phone call or post. Content of the communication was structured and consistent, regardless of the method of delivery, to ensure similar contact and information provision across participants. These communications contained recipe ideas (plant based diet group) and theory developed motivational messages to encourage continuation of the trial-group intervention”

- The primary outcome was adherence at 6 months. This is defined as the 4 weeks prior to the most recent entry in daily monitoring form. It's not clear why this definition was chosen. Why not use an adherence rate over the whole intervention period, or track weekly adherence rates?

Our primary outcome was adherence at 6 months. Thus, our point of primary interest was how well the men were adhering at the end of the intervention, as opposed to at various time points across the 6 month period. We used self-report data from the 4 weeks prior to their latest entry to allow us to see a range of data points (4 weeks), as close to the end of the intervention period as was possible (the most recent entry). We anticipated that adherence may change over time, and therefore did not want to take an average across the 6 month period. We feel that the data chosen to represent 6 month adherence was the best approximation of this outcome.

After consideration of the reviewers comments, we have added a plot (Figure 2) to illustrate the adherence trajectories for the intervention groups across the intervention duration. We have also added descriptive text to the methods (line 266) and results (line 299).

Results

- The results indicate that men in the plant-based diet intervention were consuming more fruit and vegetables than men in the control group. Are you able to determine whether this is as a result of increasing fruit or vegetables or both?

The consumption of fruit and vegetables reported here was calculated using self-reported daily portions of fruit and vegetables, this was not broken down further into specific fruit or vegetables, therefore we cannot provide any further information here.

- Please report how energy expenditure was assessed? Based on the secondary outcomes measures in the method, I had expected to see self-reported physical activity behaviour reported here instead of energy expenditure.

Energy expenditure was estimated using the RPAQ which is a valid tool for estimating total energy expenditure. It was felt that this was the most suitable outcome to report within the scope of the paper.

- For the exploratory analysis of change in adherence, I would like to see the results quantify the change from baseline to 6 months.

We thank you for this interesting suggestion, and as a result, and following consideration of a previous comment, we have added a plot of the adherence trajectories across the duration of the intervention for each intervention group (Figure 2). We have also added descriptive text to the methods (line 266) and results (line 299).

- To determine the effect of the intervention communications, it would be interesting to see if adherence rates were stable over the intervention period, or if adherence increased with the scheduled study communications.

In addition to the plots created in response to the previous comment (see Figure 2), we have additionally overlaid the communication timepoints onto the trajectories of adherence, to see if any patterns are evident. Care should be taken however not to place too much emphasis upon this as this was not a primary or secondary outcome.

Additional comments:

- Reference 1 quoting prostate cancer statistics in the UK is from 2016. Is there any more recent data available?

We have updated the reference, and the statistics have changed slightly. These have been amended in the text below.

“There were 47 740 new cases diagnosed in 2016 (1), accounting for 26% of UK male cancers. Furthermore, in 2016 there were 11 600 prostate cancer deaths, making the disease the second commonest cause of UK cancer-related death (1).” (line 80).

- Table 4: “Non-dairy milk consumption” is in the title but results are not reported in the table?

We have changed the title to 'Table 4: Effects of interventions on blood biomarkers at six months'

- Is there any information on the number of patients with localised prostate cancer in the UK, and the proportion treated with prostatectomy? Is prostatectomy the recommended treatment for men diagnosed with localised prostate cancer?

Approximately 60% of prostate cancer diagnosis in the UK are localised (stage I and stage II), and approximately 39% would have radical prostatectomy. It is therefore a common treatment for localised disease. The following text has been added to the manuscript.

"Radical prostatectomy (surgical removal of the prostate) is a common treatment for localised disease (1)." (line 113)

- What is the evidence for choosing an adherence rate of more than 90%? Has this threshold been used in other studies?

In the manuscript we state that we anticipate an intervention adherence rate of 75% (line 205). We define this adherence as the individual carrying out the instructed intervention for 90% or more of the time 'the threshold of acceptable adherence' (line 222). For clarity we have added the following text to the manuscript.

"Our intended adherence rate was 75% or above." (line 216)

The following paragraph, in the original manuscript discusses the intended adherence rate of 75%

"The plant-based adherence rates were lower than the anticipated 75%, although adherence to the reduced dairy milk component were higher, almost reaching this target; the lycopene adherence rates were higher than the proposed 75%. Almost 54% of participants adhered to the physical activity intervention at six months, slightly lower than the 59% adhering within the first four weeks of the trial." (line 403)

- It's not clear why adherence to the fruit and vegetable intervention is low, given that consumption is relatively high.

Adherence to the intervention was based on the number of days that men self-reported that they had consumed 5+ portions of fruit or vegetables. Consumption of fruit and vegetables were measured

using the Food Frequency Questionnaire, which can be prone to overestimation. It's possible, and more likely, that men over-reported their consumption rather than under-reported their adherence. The following text has been added to the discussion.

“Finally, there may have been some over-reporting of fruit and vegetable consumption, either due to social desirability, inconsistent recall or as a result of previously reported issues with the FFQ (30), this may explain the disparity between the reported high consumption of fruit and vegetables, but lower adherence to the plant-based intervention.” (line 435).

- The conclusion states that the interventions were acceptable, although this specific data is not reported in this manuscript.

We agree that although a qualitative study, published elsewhere, concluded that the participants reported the intervention to be highly acceptable, we do not describe this data here. The qualitative work is referenced in the results section; however, this has been removed from the conclusion as it does not directly relate to the reported data. We have amended the first line of the conclusion as follows.

“The PrEvENT interventions were feasible as evidenced by high randomisation rates, high retention rates and excellent tolerability.” (line 463)

Reviewer: 2

Introduction

Line 89 - The authors recent review identified a variety of diets that may have a favorable impact on prostate cancer progression and mortality. Perhaps somewhere here or elsewhere in the manuscript, a rationale as to why the specific plant-based intervention was chosen. For example, men with prostate cancer, particularly those who received androgen deprivation therapy, at a heightened risk of muscle loss. Consequently, an argument could be made for the investigation of protein and/or resistance training as opposed to plant-based diet and walking.

We agree that findings from our systematic review showed some promising diet and physical activity interventions, however there were many limitations of these studies, including often being of low quality (with high or unclear risk of bias), underpowered, inadequately reported and of short duration. We therefore wanted to conduct an RCT, overcoming a number of these limitations, yet with an intervention that, if successful, could be implemented in 'real life' in men with localised prostate cancer undergoing radical prostatectomy. We have therefore added the following text after discussion of this systematic review.

“We therefore wanted to conduct a high quality, low risk of bias, adequately powered, well reported, six month trial to explore the feasibility of a simple, easy to implement ‘real life’ dietary and physical activity intervention.” (line 95)

Methods

Line 135 - It's likely that 6-weeks post-surgery was chosen to allow to for recovery from surgery prior to physical activity? With many pre-post exercise interventions in exercise oncology investigating shorter timelines of exercise in relation to surgery, or even throughout, it might be helpful to have a rationale here as to why 6-weeks post-surgery was chosen.

As suggested by the reviewer, six weeks post-surgery was chosen as the time point to introduce a physical activity intervention, as this tends to be when men are advised by their clinical team to increase levels of physical activity following radical prostatectomy. The following text has been added to the manuscript.

“Six weeks post-surgery was identified as the ideal time to approach men to participate in the trial, as this is the time point where men are generally informed by their clinical team that they should be able to begin exercising again after radical prostatectomy.” (line 144)

Lines 150-159 - was any other dietary advice offered beyond consumption of fruit and vegetables? Was calorie or protein intake discussed at all? Given the impact of PC treatment on weight and lean body mass loss, these are certainly of concern in this population. It might be worth providing more detail of the intervention here.

Dietary advice was provided as per protocol, so men in the plant-based diet group were instructed to eat as many portions of fruit and

vegetables as possible a day, aiming for at least 5 daily portions (printed instructions explained how to measure a portion). In addition, men in the plant-based diet group were asked to reduce their dairy milk intake as much as possible and to use a non-dairy alternative, for example soy, almond or rice milk as often as possible (line 159). No other dietary advice was provided. If men in any of the groups requested further advice, they were provided with publicly available information (line 170).

Further, how were men advised on the physical activity/nutrition guidelines - was this a one day event where men were given an overview. Was it group or individual discussions? At what point did the men receive this information? Any additional information here would be great for interpretation!

All intervention instructions, for example, how to carry out brisk walking, or how many portions of fruit and vegetables to consume, were provided by the research nurse, orally and in printed format, at the time of randomisation. The additional text below has been added to the manuscript.

“Men were randomised at the trial baseline appointment, where they were provided with all information related to their intervention group in person by the research nurse.” (line 154)

Lines 161-162 - It would make sense that varying methods of communication were chosen to keep in touch with participants. Was there any rationale as to which form of communication was chosen (participant preference for example), or if a combination was used with each participant. It would be reasonable to suggest that different modes of contact may have different degrees of impact vs. efficacy (i.e. 30 min phone call vs letter in the post). Some further insight into decisions made here would be helpful in the interpretation of this intervention.

As stated in the original manuscript (line 174), the method of communication (for the majority of these communications) was chosen by the participant. In some instances, these needed to be in paper format, for example the sending of recipes for the men in the plant-based group. The content of these communications was similar, regardless of the method. For example, the text within the email, letter and sms message was exactly the same. Where a phone call was made, the caller followed a script, which was based on the content of the other communications to keep the information provided similar across all participants. To clarify this, the following text has been added to the manuscript.

“Content of the communication was structured and consistent, regardless of the method of delivery, to ensure similar contact and information provision across participants.” (line 174)

Participant Characteristics

- Do the authors have any further information on treatment participants received for their diagnosis? Did anyone received chemotherapy, radiation or hormonal therapy whilst participating in the intervention?

To be eligible to participate in the trial, participants had to have localised prostate cancer and be booked in to under radical prostatectomy at time of recruitment. This is one of the standard treatment pathways for localised disease, and it would be unusual for further treatment to be provided, unless disease progressed at a later stage. We did not collect information about any further treatment once men had been recruited into the trial. Therefore, we do not have any additional information about additional treatment.

Table 1. It might be helpful to include baseline measures of nutrition and PA in the tables where effects are displayed. It is difficult to go back and forth between tables when examining baseline values vs 6-month data.

Each of the tables already contains quite a lot of information. We feel that presenting baseline values of fruit and vegetable consumption, non-dairy milk consumption and total energy expenditure alongside their six-month equivalents in Table 3 would make the comparison of trial arms at six months less clear.

How was energy expenditure calculated? Was this extrapolated from PA questionnaires or accelerometry data? This information would be valuable to include in the methods

Energy expenditure was extrapolated from the RPAQ questionnaires. The following text is in the original manuscript.

“Physical activity energy expenditure was derived from the RPAQ” (line 246)

Could the authors estimate overall calories and macronutrient information? If a definitive RCT is being proposed from the results of this trial, this information would be critical in determining if compensatory dietary patterns were formed as a result of advice from different interventions.

This is an interesting question, however this study would be underpowered to estimate this from the available data and it would sit outside the scope of this feasibility results paper.

Is there a reason other timepoints of data collection aren't included? This would be extremely valuable in tracking timeline of intervention effects.

Our primary outcome was adherence to the intervention at six months, therefore the original manuscript focuses on this. However, following consideration of this comment, and suggestions made by Reviewer 1, we have added a plot to illustrate the trajectories of adherence across the duration of the intervention for the different intervention groups (Figure 2). We have also added descriptive text to the methods (line 266) and results (line 299).

Line 312 it's unclear looking at the data in the tables for increases in these micronutrients, what the clinical relevance of these small increases in intake are (e.g. 1.35 umol/l v 1.2 umol/l v 1.20 umol/l for lycopene). A discussion of this would be helpful.

As a feasibility trial, our outcomes were not clinical relevance regarding changes in lycopene level, or macronutrients relating to fruit and vegetable consumption. We were interested in whether men were happy to take the supplements, or make changes to their dietary behaviours, and adhere to these changes over the intervention period. To clarify this, we have added the following text to the discussion.

“It should be noted that as a feasibility trial we were not concerned with the clinical relevance of these changes, but whether participants would make, and adhere to, the changes for the duration of the trial.” (line 410)

Discussion

One concern regarding the interpretation of the result of this trial is regarding the effects of the plant based recommendations for increasing fruits and vegetables. All groups were consuming greater than 5 servings per day at baseline. Consequently, it's unknown if there is a benefit to increasing intake beyond this. Certainly, it might be helpful to discuss the implications of this more. Playing extreme devils advocate, one could question the value of a nutrition intervention aimed at increasing fruit and vegetable consumption if individuals were already consuming an adequate amount?

This is a very interesting observation. It would however have been a logistical challenge to screen for, and only recruit, men who do not eat '5-a-day'. There is some uncertainty within the literature around whether '5-a-day' guidance is clinically relevant. Therefore, we encouraged those eating the recommended daily intake (RDI) to consume more fruit and vegetables, and those not meeting the RDI to try and meet the RDI of '5-a-day'. As a feasibility trial, we also wanted to see whether a combination of '5-a-day' plus reduction in dairy milk, in addition to a physical activity intervention was feasible. We therefore have not added further text to the manuscript here.

The authors do a great job of offering insight as to why the PA intervention was largely ineffective at increasing energy expenditure. What might be helpful to discuss would be 1) why EE is chosen as the outcome for the physical activity intervention - Is this following the line that increases in EE could be helpful in weight management? If so, in addition to discussing the potential need for more intense PA interventions, could be the discussion of interventions that would be targeting to clinically relevant outcomes in this patient population (loss of lean body mass, loss of bone mineral density and accumulation of fat mass from various treatments). Certainly, it appears as though a lifestyle intervention aimed at weight management rather than PA or EE alone would be valuable here.

This is an interesting point to consider. Energy expenditure, for the purpose of this feasibility trial was considered a method of measuring how effective the intervention was at promoting moderate or vigorous physical activity. With regards to weight loss, for the purpose of the feasibility trial, this was not an outcome of interest. It is however an interesting area; and can have an impact upon recovery from surgery and overall health and quality of life. This may especially be the case within other treatment groups, for example those undergoing ADT, where weight management could mitigate some of the side effects of treatment. However, for the purpose of the feasibility trial, we feel that exploring this further is outside the scope of the paper. The follow text has been added to the manuscript.

“with the potential to influence weight management.” (line 446)

A primary concern reading this manuscript is the adherence to various physical activity and nutrition interventions as an outcome (as they relate to the specific interventions selected). For example, if a definitive RCT is proposed from the results of this trial - is an intervention aimed at increasing brisk walking and fruit and vegetable consumption sufficient to alter clinically relevant outcomes? Certainly, it appears as though both the PA and nutrition interventions, whilst adherence was high, didn't have a

meaningful impact PA levels, EE, or nutritional intake (beyond a slight increase in fruit and vegetable intake) It might be helpful to have a discussion surrounding how future research might build on this trial. For example, the physiological and psychosocial burden of prostate cancer treatments are well documented in the literature. Further, there is a wide variety of exercise and nutrition interventions aimed at either weight management or lean body mass, bone health or other outcomes impacted by cancer treatment. It is unclear how expanding this intervention or moving forward with a definitive RCT would lead to improvements in health-related outcomes in this population.

This is another interesting discussion point, however we feel it is important to highlight that as a feasibility trial we were not looking to find meaningful change in physical activity, energy expenditure or nutritional intake; these were described for completeness of the data. Our primary outcomes were randomisation rates and adherence to the intervention. It is a valid point however that our physical activity intervention for example may not have been intensive enough to see improvements in health related outcomes and would need to be developed further prior to implementing it in a full RCT. This has been discussed and acknowledged in the discussion, line 441.

VERSION 2 – REVIEW

REVIEWER	Fiona Stacey The University of Newcastle, Australia
REVIEW RETURNED	30-Jul-2019

GENERAL COMMENTS	Thank you for addressing the reviewer comments. I am satisfied that all comments have been addressed and I have no further comments.
--

REVIEWER	Ciaran Fairman Edith Cowan University
REVIEW RETURNED	10-Aug-2019

GENERAL COMMENTS	The authors have addressed all of my comments from the first round of revisions. Nothing more to add.
---